# Cite-While-You-Generate: Training-Free Evidence Attribution for Multimodal Clinical Summarization

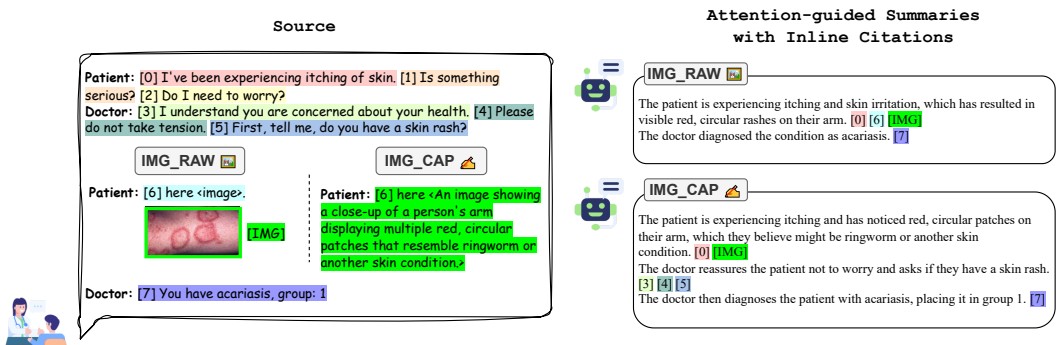

Figure 1: Overview of our proposed framework for inference-time source attribution in clinical summarization. Given a dialogue and an associated image, our method produces summaries with explicit citations to supporting evidence. We support two modes within the same framework: in the IMG_RAW mode, decoder attentions over image patches are directly aggregated to attribute visual evidence, whereas in the IMG_CAP mode, the image is converted into a one-sentence caption to enable purely text-based alignment. Each generated summary sentence is annotated with citations to source text spans and/or the image, providing interpretable and trustworthy clinical summaries.

## Abstract

Trustworthy clinical summarization requires not only fluent generation but also transparency about where each statement comes from. We propose a training-free framework for *generation-time* source attribution that leverages decoder attentions to directly cite supporting text spans or images, overcoming the limitations of post-hoc or retraining-based methods. We introduce two strategies for multimodal attribution: a raw image mode, which directly uses image patch attentions, and a caption-as-span mode, which substitutes images with generated captions to enable purely text-based alignment. Evaluations on two representative domains: clinician-patient dialogues (CliConSummation) and radiology reports (MIMIC-CXR), show that our approach consistently outperforms embedding-based and self-attention baselines, improving both text-level and multimodal attribution accuracy (e.g., +15% F1 over embedding baselines). Caption-based attribution achieves competitive performance with raw-image attention while being more lightweight and practical. These findings highlight attention-guided attribution as a promising step toward interpretable and deployable clinical summarization systems.

## 1 Introduction

Multimodal large language models (MLLMs) are increasingly applied to clinical summarization tasks, where long dialogues or imaging reports must be distilled into concise, actionable notes. These systems promise to reduce documentation burden, improve continuity of care, and accelerate

information retrieval for clinicians. Recent advances in instruction-tuned models have yielded highly fluent summaries, often rivaling or surpassing supervised baselines on automatic metrics.

However, fluency is not enough. MLLMs remain prone to *hallucinations*: claims that are weakly supported or unsupported by the underlying evidence. In high-stakes domains such as radiology and patient documentation, this problem is particularly acute: an unsupported statement about a lesion or a misattributed patient complaint can lead to delayed diagnosis, unnecessary tests, or erosion of clinician trust. For real-world adoption, practitioners require not only concise summaries but also *traceability*: a clear indication of where each statement originated in the source material. Transparency is thus a prerequisite for safe deployment of summarization models in healthcare.

Unfortunately, existing approaches provide limited solutions. Current clinical summarization systems typically output text-only summaries without citations. Post-hoc interpretability tools (Xu et al., 2023; Agarwal et al., 2022) can highlight relevant spans or patches, but they often operate at coarse granularity, do not align well with human-interpretable units such as sentences, and require additional models or retraining that are impractical in hospital workflows. Embedding-based retrieval (Karpukhin et al., 2020; Khattab & Zaharia, 2020) can approximate alignments, yet such methods conflate semantic similarity with factual grounding and struggle when visual evidence is involved. As a result, there remains a gap between what MLLMs produce and the kind of accountable, expect.

We address this gap with a **training-free, attention-guided attribution framework** that produces sentence- and image-level citations *during generation*. Our approach leverages decoder attention tensors to map generated tokens back to their most influential source spans, thereby constructing a structured citation map in real time. Unlike post-hoc explanations, this design integrates attribution directly into the generative process, ensuring that evidence is available alongside every generated statement.

As illustrated in Figure 1, we explore two complementary modes for multimodal inputs: (i) *raw image attribution*, which links output tokens directly to image patch embeddings, enabling faithful visual grounding; and (ii) *caption-as-source attribution*, which substitutes model-generated captions for image placeholders, allowing the system to align purely in text space when raw images are unavailable or impractical to share. Together, these modes support both high-fidelity grounding and lightweight deployment.

We evaluate our method in two representative clinical domains: (i) doctor–patient dialogues (CLICONSUMMATION (Tiwari et al., 2023)), which include both text-only encounters and cases with embedded image references, enabling evaluation of sentence- and image-level attribution in multimodal summarization; and (ii) radiology reports (MIMIC-CXR (Johnson et al., 2019)), where we focus on the FINDINGS → IMPRESSION summarization task, a standard benchmark for clinical reasoning with free-text inputs.

Using two state-of-the-art open MLLMs Qwen2.5-VL (Bai et al., 2025) and LLaVA-NeXT (Liu et al., 2024), we benchmark our framework against strong baselines including embedding-based similarity and model self-attention. Across both domains, our method yields higher precision, stronger exact-match performance, and more robust handling of multimodal evidence. Beyond numerical gains, our approach highlights how attention distributions, often dismissed as noisy, can be systematically aggregated into clinically meaningful attributions.

**Our contributions are threefold:**

1. We propose the first training-free method that produces *fine-grained multimodal citations during generation*, aligning output sentences with supporting text spans and images.

2. We introduce two complementary strategies for visual attribution: raw image mode and caption-as-source mode, highlighting trade-offs between fidelity and efficiency.

3. We demonstrate consistent improvements over strong baselines on two clinical summarization benchmarks, providing insights into how attention-guided attribution can enhance the trustworthiness, transparency, and deployability of MLLMs in healthcare.

## 2 RELATED WORK

**Source Attribution** Early work on faithfulness for abstractive summarization highlighted how models often produce unsupported content and motivated evidence-grounded evaluation and training signals. Representative evaluation frameworks include SummaC (Laban et al., 2022), which compares summary–source entailment with NLI-style consistency checks, and downstream factuality probes such as SelfCheckGPT (Manakul et al., 2023) that estimate hallucination risk via self-consistency checks. Directly linking generated text to its provenance has gained traction. Traceable Text (Kambhamettu et al., 2024) introduces phrase-level provenance links that attach source spans to generated spans, enabling fine-grained, inspectable justifications. Closer to our setting, Suhara & Alikaniotis (2024) formulates *source identification* for abstractive summarization, assigning each generated sentence to supporting source sentences, which we extend to the multimodal case with fine-grained sentence-level attributions. Beyond summarization, causal tracing work such as Meng et al. (2022) identifies neuron activations that are decisive in GPT models' factual predictions by corrupting inputs or altering internal activations, and demonstrates how factual associations can be edited once their storage locations in the network are identified. This line of research underscores the broader importance of linking outputs back to internal mechanisms or provenance signals, which we operationalize here via attention-guided attribution.

**Summarization in Clinical Settings** Radiology and broader clinical summarization raise uniquely high-stakes requirements for faithfulness and transparency. Surveys of radiology report generation/-summarization (Sloan et al., 2024) emphasize persistent gaps in factual grounding and clinical utility, calling for methods that surface supporting evidence and reduce hallucinations. Task-driven work in radiology (e.g., pragmatic report generation that conditions on indications/prior studies (Nguyen et al., 2023)) demonstrates that integrating structured context can improve clinical relevance, but interpretability often remains post hoc or coarse-grained. Beyond X-ray, the clinical NLP community has developed entity/relation resources, such as RadGraph (Jain et al., 2021), to enable the structured evaluation of factual correctness. Additionally, DocLens (Xie et al., 2024) includes attribution as one of the dimensions for evaluation.

**Multimodal Summarization** Multimodal summarization encodes visual inputs with CNN/ViT backbones and fuses them with language via cross-modal attention or gated fusion; however, many systems empirically under-weight images compared to accompanying text. Classic datasets and benchmarks include MSMO (Zhu et al., 2018) and VMSMO (Li et al., 2020), which catalyzed methods that align salient visual regions with summary content. Topic-aware modeling and improved cross-modal attention have been explored to better select image evidence (Mukherjee et al., 2022). Recent work explicitly argues for giving images a stronger influence during summarization: Xiao et al. (2025) emphasizes image-oriented objectives to mitigate text dominance in multimodal summaries. In contrast to these approaches, our goal is complementary: we *instrument* open multimodal LLMs to expose attentions, aggregating them into sentence- and image-level attributions that can be visualized and evaluated, thereby providing provenance signals during summarization rather than only post-hoc checks.

## 3 METHOD

We present an attention-guided attribution framework that produces fine-grained citations during summarization. Our design balances three goals: *fidelity* (faithfully mapping generated statements to evidence), *efficiency* (training-free inference without auxiliary models), and *generality* (supporting both text-only and multimodal inputs among different models).

At its core, the method is a **single pipeline** that operates on any input: a clinical dialogue, a radiology report, or a multimodal input with images. The pipeline is training-free and leverages cross-attention weights that are already available in decoder-only transformers to construct sentence-level and image-level attributions. Text-only and multimodal cases are handled through the same three-stage process, with only the treatment of visual tokens differing between modes.

## 3.1 PROBLEM SETUP

Here, we define our task of clinical summarization with source attribution. Given a source document $D$, which may be *text-only* or *multimodal*, and a generated summary $G$, the goal is to attribute each summary sentence $g_j \in G$ to one or more evidence units in $D$. For text-only inputs, evidence units are source sentences $\{s_1, \ldots, s_{|S|}\}$. For multimodal inputs, the evidence set is augmented with an image element $s_{\text{img}}$. The output is a mapping:

$$\mathcal{A} : \{1, \ldots, |G|\} \; \rightarrow \; 2^{\{1,\ldots,|S|\} \cup \{s_{\text{img}}\}},$$

which specifies, for each generated sentence index $j$, the subset of source sentences (and possibly the image) that support it.

## 3.2 CORE PIPELINE

The central challenge in attention-based attribution is how to turn noisy, token-level attention weights into stable and interpretable citations. Raw attention distributions vary across layers, heads, and even tokens within the same sentence, so a direct visualization is not reliable. Our key mechanism addresses this by defining a principled mapping: every generated token is assigned to top-$k$ source sentences through majority voting, and sentence-level attributions are then aggregated through normalized thresholding.

Formally, let $\bar{\mathbf{a}}_t \in \mathbb{R}^K$ denote the pooled attention distribution from a generated token $y_t$ to all source tokens $\{x_1, \ldots, x_K\}$. For each token $y_t$, we collect the top-$k$ attended source tokens, map them to their corresponding sentence IDs, and assign the token to the majority-vote sentence label: $\hat{s}(t)$. For a generated sentence $g_j$ with tokens $T_j$, we then aggregate token labels with a normalizing threshold $\tau$:

$$\mathcal{A}(j) = \Big\{ i : \Big( \tfrac{1}{|T_j|} \sum_{t \in T_j} I[\hat{s}(t) = i] \Big) \geq \tau \Big\}.$$

This yields the set of supporting sources $\mathcal{A}(j)$, ensuring that each generated sentence is attributed only to input sentences (or image spans) that receive consistent support across its tokens.

Our approach is deliberately training-free, requiring no parameter updates or auxiliary models. Instead, it leverages information that every transformer-based decoder already produces: cross-attention weights between generated tokens and source tokens. To transform these raw weights into meaningful citations, we follow a three-stage pipeline common to all input modes: (TEXT, IMG_RAW, IMG_CAP).

1. **Chunking.** The first step is to segment the source document into units of evidence that align with how clinicians reason about information. For text-only inputs, this means splitting the document into sentences and assigning each source token to the sentence it belongs to. For multimodal cases, we extend the evidence set by either retaining image placeholders (IMG_RAW) or replacing them with textual captions (IMG_CAP). This ensures that later stages of the pipeline can operate at sentence-level granularity, which is more interpretable to clinicians than token-level alignment.

2. **Attention pooling.** Transformers produce a large number of attention heads and layers, each of which may highlight different parts of the input. Rather than relying on a single head, which can be unstable, we average attention scores across all layers and heads for each generated token. This pooling yields a robust estimate of how strongly each output token attends to each input token. Importantly, it also ensures that our framework can be applied to different models without hand-tuning, since the averaging step normalizes away idiosyncrasies of specific architectures.

3. **Mapping and aggregation.** The final stage maps token-level attributions into sentence-level citations and filters out spurious alignments. For every generated sentence, we aggregate the source labels of its tokens and apply a threshold $\tau$ to decide which sources are retained. This prevents noise from a few stray tokens from being misinterpreted as evidence and guarantees that only sources with consistent support are considered citations. From a clinical perspective, this filtering step is crucial: it avoids overwhelming users with false positives and ensures that every citation shown in the final summary reflects a substantial alignment between the generated content and the source material.

---

**Data** : $\mathcal{D}$: items with source $S$ and optional image IMG
**Input** : model $M$ (output_attentions = True), processor $P$, mode $\in \{$TEXT, IMG_RAW, IMG_CAP$\}$, top-$k$, thresholds $\tau$
**Output** : Per-sentence citations $\mathcal{A}$ and summaries $G$

**foreach** *item* $(S, \text{IMG}) \in \mathcal{D}$ **do**

    /* --- Prompting and generation --- */
    **if** *IMG_RAW* **then** prompt $\leftarrow$ (image, $S$);
    **else if** *IMG_CAP* **then** $c \leftarrow M.\text{caption}(\text{IMG})$;
    ;
    replace $<\text{image}>$ in $S$ by $\langle c \rangle$; prompt $\leftarrow S$;
    **else** prompt $\leftarrow S$ // text-only inputs;
    $(G, \{\mathbf{A}_t\}) \leftarrow M.\text{generate}(P(\text{prompt}), \text{greedy}, \text{attn} = \text{"eager"})$;

    /* --- Chunking and alignment --- */
    $\mathcal{C}_S \leftarrow \text{SentenceChunks}(S)$, **if** *IMG_RAW* **then** record image token block $I$; shift indices in $\mathcal{C}_S$ past $I$;
    $\mathcal{C}_G \leftarrow \text{SentenceChunks}(G)$;
    **if** *IMG_CAP* **then** $i^\star \leftarrow \text{CaptionSid}(S, \mathcal{C}_S, c)$;

    /* --- Generated Token to source (pooled attention) --- */
    **foreach** *generated token* $y_t$ **do**
        $\bar{\mathbf{a}}_t \leftarrow \text{mean}_{\ell,h,q}(\mathbf{A}_t^{(\ell,h)}[q,:])$;
        ; // $\ell$: average over layer, head, query to get a single attention vector over source tokens
        $\hat{s}(t) \leftarrow \text{majority}(\text{topk\_text}(\bar{\mathbf{a}}_t, k) \rightarrow \text{sentID})$;
        **if** *IMG_RAW* **then** $w_{\text{img}}(t) \leftarrow \text{mean}(\bar{\mathbf{a}}_t[I])$;
    **end**

    /* --- Sentence aggregation --- */
    **foreach** *summary sentence* $g_j$ *with token set* $T_j$ **do**
        $\mathcal{A}(j) \leftarrow \{ i : \#\{t \in T_j : \hat{s}(t) = i\} \geq \lceil \tau |T_j| \rceil \}$;
        **if** *IMG_RAW* **then**
            compute $W_{\text{img}}(j) = \frac{1}{|T_j|} \sum_{t \in T_j} w_{\text{img}}(t)$ for each $g_j$;
            add IMG to sentence $g_{j^\star}$ with $\max_j W_{\text{img}}(j)$
        **end**
        **if** *IMG_CAP* **then** replace any $i^\star \in \mathcal{A}(j)$ by IMG;
    **end**
**end**

---

**Algorithm 1:** Generation-time Source Attribution

## 3.3 MODES OF ATTRIBUTION

While the three-stage pipeline defines a general recipe, the treatment of visual information requires distinct strategies. We therefore design two complementary modes that trade off direct fidelity to visual evidence against efficiency and text-only compatibility.

**Text-only.** The pipeline begins by splitting the source text into sentences and aligning tokens with their corresponding sentence IDs. During generation, each output token attends to the input sequence; we pool attentions and assign each token to its most attended source sentence. Sentence-level attribution is obtained by majority vote across tokens in the summary sentence. Details of ablation on different pipeline designs are discussed in Section 5.2.1.

**Raw multimodal (IMG_RAW).** For inputs containing both text and images, the source sequence includes a contiguous block of image tokens (one per merged patch). For each generated token, we sum its attention over all image tokens to obtain an image attribution score. At the sentence level,

we average these scores across all tokens in the sentence. The image is then attributed only to the summary sentence with the highest average image attention, in addition to any text spans.

**Caption-as-image-span (IMG_CAP).** As an alternative, we convert multimodal input to text-only by replacing each image placeholder with a one-sentence caption generated by the model. The caption span is treated as part of the source text during chunking. During inference, if a summary sentence is attributed to the caption span, we interpret this as an image citation. This mode removes the need for direct patch-level attention while preserving a textual trace to the visual evidence.

## 4 EXPERIMENTAL SETUP

### 4.1 DATASETS

We evaluate on two representative clinical domains (Table 1): (i) CLICONSUMMATION (Tiwari et al., 2023), consisting of doctor–patient dialogues with both text-only and multimodal cases; and (ii) MIMIC-CXR (Johnson et al., 2019), where we use only the FINDINGS→IMPRESSION summarization task from textual reports after filtering. The final testbed is balanced across modalities, with 256 text-only and 263 multimodal examples, totaling 519.

Table 1: **Dataset statistics.** Breakdown of test sets used for evaluation. Examples and filtering criteria in Appendix B.

| Dataset | Domain | Selection | #Test Cases |
|---|---|---|---|
| CLICONSUMMATION | Doctor–patient dialogues | 100 text-only + 263 multimodal | 363 |
| MIMIC-CXR | Radiology reports | Findings → Impression (filtered) | 156 |
| **Total** | | 256 text-only + 263 multimodal | **519** |

### 4.2 BASELINES AND METRICS

All experiments are conducted with `Qwen2.5-VL-7B-Instruct` (Bai et al., 2025) and `LLaVA-NeXT` (Liu et al., 2024), two state-of-the-art open multimodal large language models. We enable attention outputs during generation in order to compute attribution without performing any additional training or fine-tuning.

We compare against two strong non-attention baselines:

**Embedding similarity** State-of-the-art sentence-BERT (Reimers & Gurevych, 2019) (`all-MiniLM-L6-v2`) is used for text–text similarity and CLIP (Radford et al., 2021) (`ViT-B-32`) for text–image similarity. Each generated sentence is attributed to the most similar source sentences through thresholding, and to the image where similarity is maximal (see Appendix C for detailed implementation and hyperparameter tuning).

**Model self-attribution.** The same model that produces the summary is prompted to also generate citations, following the evaluation setup used for generating ground-truth labels.

We evaluate attribution quality along three dimensions: (i) **Text**: macro-averaged F1 and exact match of predicted vs. reference source sets; (ii) **Image**: accuracy of correctly citing the image; and (iii) **Joint**: exact match requiring both text and image elements to align.

### 4.3 IMPLEMENTATION DETAILS

Since gold-standard citations are unavailable, we construct a reference set using an LLM judge following Xie et al. (2024). For each generated sentence, the judge identifies the supporting source sentences (and image when present). This provides consistent supervision across systems while avoiding bias toward our own method. Prompt details are in Appendix E Hyperparameter choices for our attention-based pipeline (e.g., $k$ for top-$k$ pooling, sentence-level threshold $\tau$, image scoring rules) are fixed across all experiments.

Table 2: **Attribution results.** Text-only results use majority voting with $k=3$, $\tau=0.16$ for `Qwen2.5-VL-7B` and $k=5$, $\tau=0.12$ for `LLaVA-NeXT-7B`; multimodal results are evaluated on 163 CLICONSUMMATION samples with the same hyperparameters.

| | Method | Text-only Attribution | | Multimodal Attribution | | | |
|---|---|---|---|---|---|---|---|
| | | Macro-F1 ↑ | Exact Match ↑ | Text F1 ↑ | Text EM ↑ | Img Acc ↑ | Joint EM ↑ |
| Qwen2.5-VL | Ours (Text) | **76.33** | **58.70** | – | – | – | – |
| | Ours (IMG_RAW) | – | – | **65.52** | 30.75 | 75.47 | 26.46 |
| | Ours (IMG_CAP) | – | – | 58.37 | **38.52** | **77.85** | **34.87** |
| | Sent-Embedding | 72.14 | 41.72 | 53.21 | 10.86 | 76.53 | 8.48 |
| | Self-Attribution | 40.06 | 27.84 | 29.56 | 12.21 | 44.64 | 12.21 |
| LLaVA-NeXT | Ours (Text) | 62.28 | **23.33** | – | – | – | – |
| | Ours (IMG_RAW) | – | – | 49.37 | 13.32 | **81.49** | 18.12 |
| | Ours (IMG_CAP) | – | – | 56.31 | **28.02** | 65.13 | **21.39** |
| | Sent-Embedding | **63.85** | 13.52 | **57.11** | 25.40 | 73.77 | 17.65 |
| | Self-Attribution | 42.61 | 23.50 | 18.56 | 6.87 | 35.44 | 6.43 |

## 5 RESULTS AND ANALYSIS

We now turn to empirical evaluation of our attribution framework. Beyond simply reporting numbers, our goal is to analyze whether attention-based attribution can consistently outperform strong baselines across both text-only and multimodal summarization. We also ask whether different design choices such as how many tokens to aggregate or how to treat visual evidence, affect the stability and interpretability of the resulting citations.

### 5.1 MAIN RESULTS

#### 5.1.1 TEXT-ONLY ATTRIBUTION

Table 2 reports results on the 256 text-only samples. Our attention-guided attribution substantially outperforms embedding- and generation-based baselines across both backbones.

For **Qwen2.5-VL-7B**, the best setting (majority voting, $k=3$, $\tau=0.16$) achieves **76.33** macro-F1 and **58.70** exact match, compared to 72.14 / 41.72 from Sentence-BERT and 40.06 / 27.84 from self-attribution. For **LLaVA-NeXT-7B**, our method still leads, attaining **62.28** macro-F1 and **23.33** exact match, versus 63.85 / 13.52 from Sentence-BERT. Although the margin is narrower for LLaVA-NeXT, our approach still provides stronger alignment and much higher exact match, highlighting that attention-guided attribution closes the gap between surface similarity and true grounding.

Notably, variants using "max" assignment collapse to below 13 F1 (see Table 4), underscoring that *majority voting over top-$k$ attentions is crucial* for stable attributions.

#### 5.1.2 MULTIMODAL ATTRIBUTION

Table 2 also summarizes results on the multimodal split. Both of our multimodal variants outperform all baselines across the two models.

For **Qwen2.5-VL-7B**, `IMG_RAW` achieves the strongest text grounding (65.52 F1) but weaker joint attribution, whereas `IMG_CAP` attains higher joint exact match (**34.87**) and image accuracy (**77.85**) while maintaining competitive text F1. For **LLaVA-NeXT-7B**, the pattern is consistent but less pronounced: `IMG_CAP` again yields stronger joint grounding (21.39 vs. 18.12), while `IMG_RAW` provides slightly higher standalone text F1 (49.37 vs. 56.31 for captions) and image accuracy (81.49).

This contrast highlights a practical trade-off:

- Raw patch-level attentions provide stronger standalone text grounding, but are more memory-intensive and less coherent across modalities.

- Caption-based attribution encourages more balanced evidence grounding across text and images, and is attractive for deployments where images cannot be stored or transmitted due to privacy concerns.

Baselines illustrate the limitations of alternatives. Sentence-BERT with CLIP embeddings achieves the highest image accuracy (80.53 for Qwen2.5-VL, 73.77 for LLaVA-NeXT) but fails at joint attribution (8.48 / 17.65). Self-attribution remains unreliable, performing poorly across all metrics.

**Summary**    Across both backbones and input modalities, we observe three key findings: (1) majority voting is essential for converting noisy token-level attentions into stable sentence-level attributions; (2) attention-guided methods consistently outperform embedding and self-attribution baselines across both Qwen2.5-VL and LLaVA-NeXT; and (3) multimodal captioning provides a competitive alternative to raw-image attention, improving joint grounding while reducing complexity.

## 5.2  ABLATION STUDY

To ensure stability and to highlight the upper bound of our attribution framework, we perform the ablation studies on the stronger `Qwen2.5-VL-7B` backbone. As shown in Table 2, this model consistently outperforms `LLaVA-NeXT-7B` across both text-only and multimodal attribution, making it the most suitable candidate for analyzing the impact of hyperparameter choices in detail.

### 5.2.1  HYPERPARAMETER SETTING

We ablate three hyperparameters that govern attribution stability: (1) *Top-k tokens*: the number of most-attended source tokens considered per generated token; (2) *Attribution mode*: whether the token's source sentence is chosen by the single highest-attended token (`max`) or by majority voting over the top-$k$ tokens (`majority`); (3) *Aggregation threshold* $\tau$: the minimum proportion of tokens in a generated sentence that must support a source sentence for it to be retained as attribution. Low thresholds risk spurious matches, while high thresholds may discard true but sparse evidence.

All ablations are conducted on the text-only split to isolate hyperparameter effects from multimodal variability. Figure 2 visualizes the key findings (full numeric results are provided in Appendix D).

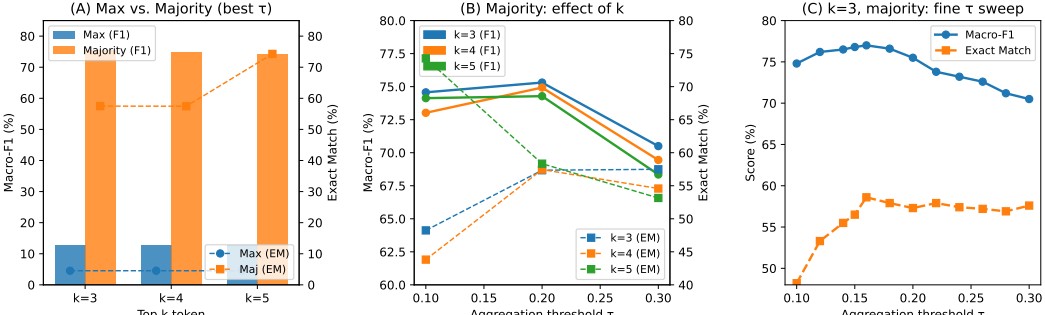

Figure 2: **Ablation on text-only attribution. (A)** Majority vs. Max (best $\tau$ per $k$): majority voting yields 60-point higher Macro-F1 and far stronger exact match than max, which collapses to near-random performance. **(B)** Effect of $k$ under majority voting: attribution quality peaks around $\tau = 0.2$, with $k = 3$ providing the best balance between robustness and stability. **(C)** Fine-grained $\tau$ sweep for $k = 3$, majority shows the optimal range lies between $0.15$ and $0.18$, supporting our choice of $\tau = 0.16$ as the default setting. Full numeric results are provided in Appendix D.

**Findings.**    Figure 2(A) contrasts `max` and `majority` modes. Across all values of $k$, `max` collapses to ~12 Macro-F1 and ~4.5 Exact Match, confirming that selecting the single strongest token is unstable and fails to capture distributed evidence. In contrast, `majority` consistently yields 60+ point gains in F1, highlighting its robustness.

Figure 2(B) explores the effect of $\tau$ under majority voting. For $k \in \{3, 4, 5\}$, performance peaks around $\tau = 0.2$: lower thresholds admit spurious matches, while higher thresholds prune true but sparse support. Although larger $k$ sometimes improves exact match (e.g., $k = 5$), it dilutes F1 and leads to less stable attribution, whereas $k = 3$ balances robustness and efficiency.

Finally, Figure 2(C) zooms in on a fine-grained sweep for $k = 3$, majority. Here, Macro-F1 reaches its maximum at $\tau = 0.15$–$0.16$ (76.4 F1), while Exact Match peaks at $\tau = 0.16$ (58.70). This validates $\tau \approx 0.18$ as a robust default.

**Choice of final hyperparameters.**   Based on these results, we adopt $k = 3$, `majority`, $\tau = 0.16$ as our default configuration. This setting achieves the best trade-off (76.33 Macro-F1 / 58.70 Exact Match) and consistently outperforms both embedding-based and self-attribution baselines.

**Implications.**   These results show that attribution stability depends not only on raw attention scores but also on how they are aggregated. Majority-based aggregation with a small top-$k$ acts as a denoising mechanism, turning noisy token-level distributions into reliable sentence-level citations. Beyond our datasets, this suggests a general recipe for applying attention-based attribution: restrict to a focused set of tokens, use majority voting to stabilize decisions, and adopt moderate thresholds to balance recall and precision. Together, these design choices transform raw attentions—often dismissed as uninterpretable—into a practical tool for generating faithful, human-auditable citations.

### 5.2.2   DOES BETTER ATTRIBUTION LEAD TO BETTER SUMMARIES?

Table 3: Summary quality evaluation with `Qwen-7B` under different multimodal settings.

| Mode | ROUGE-1 | ROUGE-L | BERTScore-F1 |
|---|---|---|---|
| IMG_RAW | 0.537 | 0.400 | 0.913 |
| IMG_CAP | 0.496 | 0.374 | 0.902 |

We further examine whether attribution quality correlates with summary quality. Table 3 reports the summary evaluation results of `Qwen-2.5-7B` under different multimodal settings, comparing raw image input against caption input.

**Analysis.**   From Table 3, using raw images yields better summary quality than captioning ($+4.1$ ROUGE-1, $+2.6$ ROUGE-L, $+1.1$ BERTScore-F1). This trend aligns with the multimodal attribution results in Table 2, where raw-image attribution achieved stronger text F1 compared to caption-based attribution. Although caption-based attribution produced higher text EM and joint EM, raw images consistently provide higher overall summary fidelity.

These findings highlight a trade-off: **Raw images** enable stronger alignment with the reference text and better overall summary metrics, at the cost of slightly lower robustness in exact span matches. **Caption input** may guide models toward more precise sentence-level matches (higher EM in Table 2), but at the expense of degraded semantic fidelity in summaries.

Overall, attribution quality does exhibit correlation with summary quality: better multimodal attribution (raw images) coincides with improved semantic summary metrics, suggesting that robust attribution mechanisms enhance the informativeness and grounding of generated summaries.

## 6   DISCUSSION & INSIGHTS

Our study yields several practical takeaways. First, in clinical summarization tasks, caption-based attribution emerges as a lightweight yet effective alternative to raw image attention. This is particularly useful in scenarios where images cannot be shared or processed directly, but textual surrogates are feasible. Second, generating source citations alongside summaries improves trustworthiness: clinicians can see not only what the model outputs, but also where each statement comes from. This aligns with broader goals of transparency and safety in clinical AI. At the same time, our approach highlights opportunities for reproducibility and deployment. Because it is training-free and model-agnostic, the method can be readily plugged into existing pipelines for auditing and validation.

Limitations remain. Attributions rely on alignment between attention weights and true evidence, and our evaluation is restricted to two datasets. Future work will extend to larger clinical corpora, explore domain-tuned models, and investigate real-world deployment in clinical workflows.

ETHICS STATEMENT

This work focuses on attribution and grounding for multimodal clinical summarization. We emphasize that our experiments are conducted on publicly available benchmark datasets (e.g., MIMIC-CXR and CliConSummation) that have undergone de-identification and institutional review, ensuring no personally identifiable patient data is used. All methods are designed for research purposes only and are not intended for direct clinical deployment without additional validation and regulatory approval. We acknowledge that automatic summarization and attribution in healthcare carries potential risks, including the possibility of generating misleading or incomplete explanations. To mitigate these risks, we stress that outputs should be interpreted as research findings rather than medical advice. We do not foresee conflicts of interest or sponsorship concerns in this work.

REPRODUCIBILITY STATEMENT

We have taken multiple steps to ensure the reproducibility of our results. All hyperparameter choices and evaluation protocols are documented in the main text (e.g. Section 5.2.1) and the Appendix (e.g. Appendix D, E). We provide complete descriptions of dataset construction and preprocessing in Appendix B. To facilitate replication, our attribution code and evaluation scripts will be made available in the supplementary materials during the review process. Further, ablation studies (Appendix D) provide insight into the sensitivity of our method to hyperparameters, and additional results across two backbones (Qwen2.5-VL and LLaVA-NeXT) demonstrate robustness. Together, these resources support faithful reproduction and extension of our findings.

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

## A    APPENDIX

## B    DATA PREPROCESSING AND FILTERING FOR MIMIC-CXR

To construct a reliable text-only test set from MIMIC-CXR, we applied several filtering steps to the raw reports:

**Step 1: Identify single-report patients.**    We first traversed the report directory and grouped all reports by patient ID. Only patients with a single report were retained to avoid complications where impressions referenced prior studies.

```
for patient_id in all_patients:
    if len(reports[patient_id]) == 1:
        keep patient_id
```

**Step 2: Extract FINDINGS and IMPRESSION sections.**    From each retained report, we extracted the `FINDINGS` and `IMPRESSION` sections using regex patterns. Reports missing either section were discarded.

**Step 3: Sentence-length filtering.**    We segmented both sections into sentences and required a minimum of $\geq 9$ sentences in `FINDINGS` and $\geq 5$ sentences in `IMPRESSION`. This ensured enough content for meaningful summarization.

```
if len(sent(FINDINGS)) >= 9
and len(sent(IMPRESSION)) >= 5:
    keep report
```

**Outcome.**    This pipeline produced **156** text-only test cases. Together with CLICONSUMMATION, our combined evaluation set is modality-balanced (256 text-only, 263 multimodal, total 519 examples).

## C    BASELINE IMPLEMENTATION DETAILS

For comparison, we implemented a simple *embedding-based attribution baseline* that relies on pretrained encoders. The method operates in two stages:

- **Text–text similarity.** Both source sentences and generated summary sentences are encoded in a text embedding space (we used a Sentence-BERT variant). Cosine similarity is computed to rank candidate source sentences for each generated sentence.

- **Text–image similarity.** When an image is present, generated sentences are additionally embedded in the CLIP text space, and compared against the CLIP embedding of the image. The image is assigned to the generated sentence with the highest text–image similarity.

Each generated sentence is then attributed to the subset of source sentences exceeding a cosine similarity threshold, together with the image if applicable. If the image is selected, it is always enforced to appear at most once per summary, assigned to the most relevant generated sentence. We experimented with different similarity thresholds and report in the main paper only the *best-performing configuration*.

Below we include the implementation used for this baseline.

```python
from sentence_transformers import SentenceTransformer
import torch, torch.nn.functional as F
from PIL import Image
from typing import List, Union

class MultimodalAttributionModel:
    def __init__(self,
        text_model_name: str = "TEXT_ENCODER", # text2text encoder
        clip_model_name: str = "CLIP_ENCODER", # text2image shared encoder
        threshold_text: float = 0.5,
        max_sources: int = 10,
        img_counts_towards_k: bool = True):
        self.text_model = SentenceTransformer(text_model_name)
        self.clip_model = SentenceTransformer(clip_model_name)
        self.threshold_text = threshold_text
        self.max_sources = max_sources
        self.img_counts_towards_k = img_counts_towards_k

    @torch.no_grad()
    def _embed_text_textspace(self, sentences):
        return self.text_model.encode(
            sentences, convert_to_tensor=True, normalize_embeddings=True)

    @torch.no_grad()
    def _embed_text_clipspace(self, sentences):
        return self.clip_model.encode(
            sentences, convert_to_tensor=True, normalize_embeddings=True)

    @torch.no_grad()
    def _embed_image_clipspace(self, image_path: str):
        img = Image.open(image_path).convert("RGB")
        emb = self.clip_model.encode([img],
            convert_to_tensor=True, normalize_embeddings=True)
        return emb[0] # [d]

    def attribute(self, source_sentences: List[str],
                generated_sentences: List[str],
                image_path: Union[str, None] = None):
        # (1) text2text similarities
        src_emb = self._embed_text_textspace(source_sentences)
        gen_emb = self._embed_text_textspace(generated_sentences)
        sim_text = F.cosine_similarity(
            gen_emb.unsqueeze(1), src_emb.unsqueeze(0), dim=-1)

        # (2) text2image similarity (CLIP), if image present
        have_img = image_path is not None
        if have_img:
            img_emb = self._embed_image_clipspace(image_path)
            gen_emb_clip = self._embed_text_clipspace(generated_sentences)
            sim_img = F.cosine_similarity(gen_emb_clip,
                    img_emb.unsqueeze(0), dim=-1)
            img_best_idx = int(torch.argmax(sim_img).item())
        else:
```

```
        img_best_idx = -1

    # (3) attribution assignment
    attributions = []
    for i in range(len(generated_sentences)):
        sim_row = sim_text[i]
        sorted_sims, sorted_idxs = torch.sort(sim_row, descending=True)

        chosen = []
        for sim, idx in zip(sorted_sims, sorted_idxs):
            if sim.item() >= self.threshold_text:
                chosen.append(int(idx.item()))
            if len(chosen) >= self.max_sources:
                break

        # enforce IMG to appear once
        if have_img and i == img_best_idx:
            if len(chosen) >= self.max_sources:
                chosen = chosen[: self.max_sources - 1]
            chosen.append("IMG")

        attributions.append(chosen)

    return attributions
```

# D    ABLATION RESULTS

Here we provide the numerical results of our ablation study in Table 4.

# E    PROMPT DETAILS FOR THE GPT ANNOTATOR

We employed GPT-based annotation to construct silver reference attributions. The same prompts were also used when implementing the *self-attribution* baseline, where the model that generated the summaries was asked to directly output source citations.

Table 4: **Ablation on text-only attribution.** Bold highlights best values.

| Top-$k$ | Attr mode | Agg. $\tau$ | Macro-F1 | Exact Match |
|---|---|---|---|---|
| 3 | max | 0.1 | 12.58 | 4.54 |
| 3 | max | 0.2 | 12.58 | 4.54 |
| 3 | max | 0.3 | 12.58 | 4.54 |
| 3 | majority | 0.10 | 74.57 | 48.24 |
| 3 | majority | 0.12 | 75.81 | 53.27 |
| 3 | majority | 0.14 | 76.18 | 55.31 |
| 3 | majority | 0.15 | **76.43** | 56.51 |
| **3** | **majority** | **0.16** | 76.33 | **58.70** |
| 3 | majority | 0.18 | 75.77 | 57.77 |
| 3 | majority | 0.20 | 75.31 | 57.35 |
| 3 | majority | 0.22 | 73.56 | 57.93 |
| 3 | majority | 0.24 | 72.97 | 57.35 |
| 3 | majority | 0.25 | 72.81 | 57.35 |
| 3 | majority | 0.26 | 72.34 | 57.18 |
| 3 | majority | 0.28 | 71.04 | 56.93 |
| 3 | majority | 0.30 | 70.50 | 57.49 |
| 4 | max | 0.1 | 12.78 | 4.54 |
| 4 | max | 0.2 | 12.78 | 4.54 |
| 4 | max | 0.3 | 12.78 | 4.54 |
| 4 | majority | 0.1 | 73.02 | 43.82 |
| 4 | majority | 0.2 | 74.93 | 57.45 |
| 4 | majority | 0.3 | 69.45 | 54.58 |
| 5 | max | 0.1 | 12.78 | 4.54 |
| 5 | max | 0.2 | 12.78 | 4.54 |
| 5 | max | 0.3 | 12.78 | 4.54 |
| 5 | majority | 0.1 | 74.13 | 74.28 |
| 5 | majority | 0.2 | 74.28 | 58.34 |
| 5 | majority | 0.3 | 68.35 | 53.15 |
| **Ours (best)** | Top-$k$=3, Majority, $\tau$=0.16 | | **76.33** | **58.70** |
| Baseline: Sent_emb | | | 72.14 | 41.72 |
| Baseline: Self_attr | | | 40.06 | 27.84 |

**Attribution Prompt (Text-only).**

You are given a list of source sentences and a list of generated sentences from the summary. For each generated sentence, identify one or more source sentences it can be attributed to based on semantic similarity and information content.

**Input Format:**

```
Source Sentences:
[0] sentence 1
[1] sentence 2
...

Generated Sentences:
[0] generated sentence 1
[1] generated sentence 2
...
```

**Output Format:**

```
[0] [source sentence ids]
[1] [source sentence ids]
...
```

**Example:**

```
Source Sentences:
[0] The patient has a fever.
[1] The patient complains of headache.

Generated Sentences:
[0] The patient is experiencing fever
and headache.

Output:
[0] [0, 1]
```

Now attribute the following:

```
Source Sentences:
{source}

Generated Sentences:
{summary}

Output:
```

**Attribution Prompt**

You are given a list of source sentences (the text contains an "¡image¿" placeholder) and one associated image. For each generated summary sentence, identify the source elements (sentences and/or image) it can be attributed to.

**Input Format:**

```
Source Sentences:
[0] source sentence 1
[1] source sentence 2
...

Generated Sentences:
[0] generated sentence 1
[1] generated sentence 2
...
```

**Output Format:**

```
[0] [source ids and/or IMG]
[1] [source ids and/or IMG]
...
```

**Example:**

```
Image shows a red eye.
Source Sentences:
[0] Doctor: Do you have eye pain?
[1] Patient: Yes, my right eye is very red.
<image>

Generated Sentences:
[0] The patient has eye redness.

Output:
[0] [1, IMG]
```

Now attribute the following:

```
Source Sentences:
{source}

Generated Sentences:
{summary}

Output:
```

