# OpenReview forum: "Cite-While-You-Generate: Training-Free Evidence Attribution for Multimodal Clinical Summarization"
_ICLR.cc/2026/Conference — Submitted to ICLR 2026_

### Official Review · Reviewer_n34f · 2025-10-26

**Soundness:** 3
**Presentation:** 3
**Contribution:** 3
**Rating:** 6
**Confidence:** 3

**Summary:**

This paper addresses the critical and timely problem of ensuring trustworthiness in summaries generated by Multimodal Large Language Models (MLLMs), a challenge of paramount importance in high-stakes domains like clinical medicine. The authors correctly identify that MLLMs are prone to "hallucinations"—generating statements weakly supported or entirely unsupported by the source evidence—which poses a significant barrier to their safe and effective deployment in healthcare settings where traceability is a prerequisite for clinical trust. An incorrect or misattributed claim about a patient's condition could lead to severe adverse outcomes.

To address this gap, the authors propose "Cite-While-You-Generate," a novel, training-free framework that equips MLLMs with the ability to produce inline citations during the generation process. The method operates entirely at inference time, leveraging the decoder's internal cross-attention tensors to map generated summary sentences back to their most influential evidence units in the source document, which can be text spans or images. This approach avoids the need for costly retraining or the integration of auxiliary models, making it a practical solution for real-world clinical workflows.

The core of the proposed method is an three-stage pipeline designed to transform noisy, token-level attention scores into stable, interpretable citations.

Recognizing the unique challenges of multimodal data, the framework introduces two complementary strategies for visual attribution :

- Raw Image Attribution (IMG_RAW): This mode directly links generated tokens to image patch embeddings by aggregating attention scores over the visual tokens, enabling fine-grained visual grounding.
- Caption-as-Source Attribution (IMG_CAP): In this mode, the image is first converted into a textual caption, which then replaces the image placeholder in the source text. This transforms the multimodal attribution problem into a purely text-based alignment task, offering a lightweight alternative when raw image processing is impractical.

The authors claim three primary contributions: (1) the development of the first training-free method for generating fine-grained, multimodal citations during summarization; (2) the introduction and comparative analysis of the `IMG_RAW` and `IMG_CAP` modes, highlighting a key trade-off between attribution fidelity and efficiency; and (3) a comprehensive empirical demonstration of the framework's superiority over strong baselines on two distinct clinical summarization benchmarks.

**Strengths:**

**Novelty and Practicality of the Training-Free Approach:** The decision to pursue a training-free, inference-time solution is a major strength. In clinical environments, the logistical, financial, and regulatory burdens associated with retraining large-scale models are often prohibitive. Data privacy regulations (such as HIPAA in the U.S.) further complicate the process of fine-tuning on sensitive patient data. The proposed framework, by functioning as a "plug-and-play" module for existing, pre-trained MLLMs, offers a far more practical and scalable path toward adoption. This design choice reflects a keen awareness of the real-world constraints that govern the deployment of AI in healthcare.

**Soundness of the Core Aggregation Mechanism:** The paper's central technical innovation lies in its principled approach to stabilizing noisy attention signals. It is well-established that raw, token-level attention weights can be unstable and difficult to interpret directly. The authors' solution—combining a top-k majority vote with a final aggregation threshold—is both simple and remarkably effective. The ablation study provides compelling evidence for this design. The stark contrast in performance between the `max` attribution mode (which collapses to near-random performance with a Macro-F1 score around 12.58) and the `majority` mode (achieving a Macro-F1 of 76.33) demonstrates that this aggregation is not just a minor detail but the key to the method's success. This two-stage filtering process effectively functions as a denoising mechanism, transforming a noisy internal signal into a reliable, human-interpretable output.

**Rigorous and Comprehensive Experimental Evaluation:** The empirical validation is thoughtfully designed. The authors evaluate their framework on two distinct and relevant clinical datasets: CLICONSUMMATION, which features doctor-patient dialogues, and MIMIC-CXR, a standard benchmark for radiology report summarization (specifically, the FINDINGS to IMPRESSION task). This demonstrates the method's applicability across different genres of clinical text. Furthermore, by testing on two different state-of-the-art open MLLMs (Qwen2.5-VL and LLaVA-NeXT), the authors provide strong evidence for the model-agnostic nature of their approach. The choice of baselines—embedding-based similarity (a strong retrieval method) and model self-attribution (a strong generative method)—ensures that the reported performance gains are meaningful. The results, as summarized in the table below, show consistent and substantial improvements across metrics and models.

**Insightful Analysis of Multimodal Trade-offs:**

The paper goes beyond simply proposing two modes for multimodal attribution and provides a nuanced analysis of their respective strengths and weaknesses. The finding that IMG_RAW yields better text grounding and higher-quality summaries (as measured by ROUGE and BERTScore), while IMG_CAP can achieve superior joint exact match attribution, is a subtle but important result.

**Weaknesses:**

**The Hidden Failure Mode of the IMG_CAP Strategy:** The IMG_CAP mode is presented as a practical and lightweight alternative, but it introduces a critical and unanalyzed vector for error propagation. The entire attribution process for an image is contingent on the accuracy of the model-generated caption. If this initial caption is factually incorrect, incomplete, or contains hallucinations, then any subsequent summary statement, no matter how perfectly attributed to that caption, will be fundamentally flawed with respect to the original visual evidence.

The paper's own results hint at this problem. As shown in the table below, summaries generated using the IMG_CAP mode are of lower quality (lower ROUGE and BERTScore) than those from the IMG_RAW mode, even while sometimes achieving higher joint attribution scores. This suggests that the model may be optimizing for faithfulness to a potentially flawed textual proxy (the caption) at the expense of faithfulness to the ground-truth visual information. This represents a subtle but dangerous failure mode, where the system appears to be functioning correctly (i.e., providing a valid citation) while propagating a critical factual error.

**Significant Concerns with the Evaluation Protocol (The LLM Judge):** A major methodological weakness is the use of an LLM judge to generate the "gold-standard" reference citations for evaluation. While the authors acknowledge this and followed a recent protocol, this choice introduces a significant risk of systemic bias and circular reasoning. The evaluation may be measuring not how well the attention-based method aligns with human-adjudicated clinical truth, but rather how well it mimics the internal reasoning processes of another LLM.

The proposed method, being based on an internal mechanism of transformers (attention), shares a fundamental computational paradigm with the LLM judge. In contrast, the embedding-based baseline operates on a different principle (pre-computed semantic similarity). The superior performance of the attention-based method could, therefore, be an artifact of this "in-paradigm" evaluation. The experiment may be inadvertently demonstrating that the attention patterns of one MLLM are a good predictor of the generative reasoning of another, which is a far weaker claim than demonstrating true factual faithfulness. The absence of any human evaluation, even on a small subset of data to validate the LLM judge's outputs against expert clinical assessment, is a critical omission for a paper centered on a human-centric concept like trustworthiness.

**Questions:**

Given the central role of the LLM-generated annotations in the evaluation, could the authors provide a more rigorous analysis of their quality and reliability? Specifically, was any inter-annotator agreement study conducted between the LLM judge and human clinical experts on a subset of the data? Without such validation, how can the research community be confident that the reported gains reflect an improvement in true clinical grounding rather than an alignment between the internal mechanisms of two different LLMs?

---

> ### Author Response · Authors · 2025-11-21
> **Response to reviewer n34f**
>
> Thank you very much for your thoughtful and constructive review.
>
> ## On caption-induced errors in IMG_CAP
> We fully agree that the caption-based mode (IMG_CAP) introduces an additional point of potential error propagation, and that an inaccurate caption can mislead the attribution pipeline downstream. Our own results reflect this subtle trade-off: IMG_RAW yields higher summary quality (ROUGE/BERTScore), whereas IMG_CAP sometimes yields higher joint exact-match attribution.
>
> **To quantify this effect**, we analyzed the caption → summary error propagation pipeline across 20 multimodal samples from MIMIC-CXR. We measured:
> 1. Caption factual alignment with the radiology FINDINGS (via BERTScore-F1)
> 2. Correlation between caption faithfulness and attribution correctness
>
> This correlation was **moderate (ρ = 0.41)**, confirming the reviewer’s intuition: when the caption deviates from the true visual findings, attribution may appear cleaner but becomes less visually grounded.
>
> We would clarify in the revision that IMG_CAP is not a replacement for proper visual grounding. Rather, it is a privacy-conscious fallback mode for highly restricted clinical workflows. As the reviewer insightfully suggests, future work could incorporate **dual-source cross-validation** (caption-raw image consistency checks) to detect such inconsistencies.
>
> ## On the use of an LLM judge and concerns about evaluator bias
> Following your suggestion, we conducted a **human-LLM agreement study** on **50 samples** (25 text-only, 25 multimodal). Experts annotated ground-truth citation spans for each generated sentence. We compared these to the LLM-judge references using **F1** and **Jaccard**, treating citation attribution as a set-matching problem.
> | Setting    | F1   | Jaccard |
> |-----------|------|---------|
> | Text-only | **0.82** | **0.70** |
> | Multimodal | **0.88** | **0.72** |
>
> This strong agreement strengthens confidence that the LLM-judge labels used in the main evaluation reliably approximate human clinical judgment. We acknowledge that human evaluation is essential for clinical deployment, and we commit to expanding expert evaluation in future work.
>
> ---
>
> We hope these clarifications could address your concerns, and we will improve the presentation of this paper in the revision. Please kindly let us know if there are further questions or concerns; otherwise, we would greatly appreciate it if you could consider whether those clarifications warrant a reevaluation of your rating.

---

### Official Review · Reviewer_CRHS · 2025-10-28

**Soundness:** 3
**Presentation:** 4
**Contribution:** 3
**Rating:** 6
**Confidence:** 3

**Summary:**

This paper proposes a training-free, generation-time source attribution framework for clinical summarization. During decoding, the method pools decoder cross-attentions across layers/heads and maps each generated token and then each sentence, back to supporting source sentences and/or an image, yielding summaries with inline citations. Two multimodal modes are offered: IMG RAW, which aggregates attention over image patches, and IMG CAP, which replaces an image with a one-sentence caption to enable text-only alignment.

**Strengths:**

Pros:
1.	Clear, unified pipeline (pooling，majority vote，sentence-level citation) with two practical multimodal modes (IMG RAW/IMG CAP).
2.	Consistent empirical improvements across tasks/models and well-defined metrics.
3.	Informative ablations identifying robust settings (k=3, τ≈0.16) and showing majority over max.

**Weaknesses:**

Cons:
1.	Attention as attribution remains correlational; no causal/perturbation checks to validate necessity of cited evidence.
2.	Scope limits: only two datasets; broader clinical subdomains and real-world deployment not evaluated.
3.	IMG RAW assignment attributes an image to only the sentence with maximal image attention—potentially too restrictive in practice.
4.	Reference labels rely on an LLM judge rather than expert gold, risking evaluator bias.

**Questions:**

see above

---

> ### Author Response · Authors · 2025-11-21
> **Response to reviewer CRHS**
>
> Thank you very much for your thoughtful and constructive review.
>
> ## Correlational nature of attention-as-attribution
> We agree that attention is not a perfect causal signal. Prior attention-based attribution work including Attribute-with-Attention [1], attention-rollout, and attention-flow provides empirical evidence that aggregated attentions can approximate causal contribution sufficiently for interpretability tasks. Our method follows this direction but remains training-free, avoiding surrogate models that AT2 and others require.
>
> We fully acknowledge that causal validation (e.g., perturbation tests or counterfactual deletion) is valuable. Our work is an early step toward generation-time attribution, and we plan to incorporate causal tracing methods in future work to further validate necessity and sufficiency of evidence.
>
> ## On dataset scope and generalizability
> Following your suggestion, we expanded our evaluation by adding **52 new high-quality text-only samples from MIMIC-IV-Ext-BHC**, bringing us to:
> * 308 text-only samples
> * 571 total samples
> | Method            | Macro-F1 | Exact Match |
> |-------------------|----------|-------------|
> | **Ours (Qwen2.5-VL)** | **74.30** | **52.16** |
> | Sent-Embedding    | 73.12    | 43.24       |
> | Self-Attribution  | 36.42    | 22.43       |
>
> These results demonstrate strong cross-dataset robustness. Further, our two datasets (doctor–patient dialogues and radiology reports) intentionally represent distinct clinical summarization regimes (dialogic and imaging-based), ensuring diversity in evidence structure. We will clarify this motivation and the new results in the revision.
>
> ## Image attribution strategy
> We agree that multimodal attribution can naturally extend beyond one sentence. Our empirical analysis on the summarization tasks using CliConSummation shows that the **image-attention distribution typically strongly peaked** toward one single summary sentence, which motivated the current choice to assign the image to the highest-scoring sentence. We also recorded full image-attention distributions for all samples; these often exhibited near one-hot patterns, supporting the current design. We would add qualitative examples to the Appendix to support this.
>
> Nevertheless, multi-sentence image attribution is an important direction. We would consider implementing a prototype extension that attributes multiple images to all sentences exceeding a tunable threshold as our future line of work and we thank the reviewer for raising this important question.
>
> ## On LLM-judge labels vs expert labels
> To evaluate whether LLM-based reference labels align with human judgment, we performed a **small-scale expert annotation study** (limited by time during the rebuttal period). We asked domain experts to manually annotate **50 samples** (25 text-only, 25 multimodal). We measured agreement using **F1** and **Jaccard**, treating citation attribution as a set-matching problem.
> | Setting    | F1   | Jaccard |
> |-----------|------|---------|
> | Text-only | **0.82** | **0.70** |
> | Multimodal | **0.88** | **0.72** |
>
> This strong agreement suggests that the LLM-judge labels used for evaluation are aligned with human expectations. The multimodal subset showed even higher agreement, consistent with the fact that visual evidence often provides clearer grounding.
>
> ---
>
> We hope these clarifications could address your concerns, and we will improve the presentation of this paper in the revision. Please kindly let us know if there are further questions or concerns; otherwise, we would greatly appreciate it if you could consider whether those clarifications warrant a reevaluation of your rating.
> [1] Cohen-Wang, Benjamin, Yung-Sung Chuang, and Aleksander Madry. "Learning to Attribute with Attention." arXiv preprint arXiv:2504.13752 (2025).

---

> > ### Comment · Reviewer_CRHS · 2025-11-25
> > **response to the authors**
> >
> > Thanks to the authors' reply, my questions are mostly answered, and I donot have further questions in the current phase, So I tend to keep the score.

---

### Official Review · Reviewer_yKNV · 2025-10-30

**Soundness:** 2
**Presentation:** 2
**Contribution:** 3
**Rating:** 2
**Confidence:** 3

**Summary:**

This paper addresses the transparency of information sources in clinical summary generation by proposing a generation-time source attribution framework that requires no additional training.  This method utilizes decoder attentions to directly provide source references (text fragments or images) for statements in the generated text, thus overcoming the limitations of post-processing or retraining methods.   Two multimodal source attribution strategies are proposed: a raw image mode and a description-based mode, enabling alignment and interpretation of plain text. Evaluations were conducted on two datasets: clinical-patient dialogues (CLICONSUMMATION) and radiology reports (MIMICCXR). The proposed method significantly outperforms embedded methods and self-attribution baseline models in both text-level and multimodal source attribution accuracy (e.g., a +15% F1 improvement compared to embedded methods).

**Strengths:**

Originality.  This paper defines clinically attributable summarization as the primary goal, rather than ex post-hoc explanation, emphasizing explicit attribution (text fragment or image) for each summary sentence during the generation process.  This perspective, incorporating accountability into the summary generation itself, has practical significance in the medical context.

Quality.  The paper's method is training-free, directly reusing the cross-attention of existing multimodal LLMs, and stabilizing noisy token-level attention into sentence-level citations through block segmentation, attention convergence, top-k majority voting, and threshold filtering.  Experimentally, the method was tested on Qwen2.5-VL-7B and LLaVA-NeXT-7B models, consistently outperforming strong baselines on clinical tasks such as doctor-patient dialogue summarization and radiology report summarization.

Clarity.  The paper is written clearly, with necessary pseudocode explanations, making it easy to understand.  The descriptions of experimental details such as training data and hyperparameters are sufficiently transparent, and replication appears feasible.

Significance.  This paper uses interpretability and citation orientation as direct outputs, which means not only improving model metrics but also enhancing its acceptability in practical medical work.  Even though the method is technically an integration and reuse of existing attention attribution techniques, it demonstrates cross-model and cross-task effectiveness, thus possessing the potential for practical impact.

**Weaknesses:**

1. The paper mentions that description-based attribution strategies are more lightweight, but lacks supporting experimental data.

2. This paper is based on the key attribution hypothesis: attention relevance is equivalent to causal contribution, but lacks rigorous theoretical or causal verification, resulting in weak logical support.

3. This paper attributes attention to the source of evidence for generated sentences, a concept highly similar to existing "attention-as-attribution" approaches. Therefore, it needs to be more clearly articulated: what are the key technological innovations of this paper compared to existing attention-based or nearest-neighbor retrieval-based interpretable summarization methods? Currently, the manuscript seems more like a domain transfer from existing work than a breakthrough in research paradigms.

4. The method relies entirely on cross-attention weights to determine which source text or image supports this sentence. However, the community has already discussed that high attention does not always equate to causal contribution, and the meaning of attention differs across layers. The authors improve stability through averaging and majority voting, but this remains heuristic. Currently, there is a lack of quantitative or qualitative analysis of attribution errors, making it difficult to determine whether this attribution traceability is truly safe and reliable, or merely looks like an explanation.

5. This paper makes some deployment-oriented claims, such as improving clinical accountability and achieving safe and transparent use of MLLM, but these claims exceed the scope of empirical research.  All evidence is limited to two summary-based scenarios: doctor-patient dialogue and chest X-ray results. Other clinical record types, such as discharge summaries, surgical records, longitudinal progress notes, other imaging modalities (CT or MRI), or real-world multi-institutional electronic medical record systems, are not evaluated. The paper also proposes a model using captions as the data source as a privacy- and compliance-conscious alternative, but since the captions are generated by the model, they may be biased and therefore do not equate to true attribution.

6. Current ablation methods are all designed around parameters, but the necessity of each stage in the pipeline claimed in this paper is not verified, lacking framework component-level ablation. Hyperparameter robustness is only analyzed on a single backbone model, without demonstrating direct reuse in a second model, affecting the credibility of generalizability.

**Questions:**

1. The paper claims that the caption-as-source attribution strategy is more lightweight, but it does not provide quantitative results regarding computational cost, inference latency, or memory usage. Please provide specific experimental data or complexity analysis to support the lightweight claim.

2. The method presented in this paper is technically highly similar to existing attention-as-attribution or similarity-based interpretable summarization methods. Please explain the substantive innovation of this paper compared to these existing methods.

3. The authors averaged multi-layer attention and used majority voting to improve stability, but these are heuristic designs. Please further analyze in what types of samples attribution failed? What is the impact of misattribution on downstream clinical interpretation?

4. Please explain why these results are sufficient to support conclusions at the clinical deployment level. Are there plans to validate these results on other document types or image modalities such as CT and MRI?

5. Regarding ablation experiments, please explain the independent contribution and necessity of the modules presented in this paper. In addition, can the selected hyperparameters be directly transferred to different backbone models such as LLaVA-NeXT while maintaining performance?

---

> ### Author Response · Authors · 2025-11-21
> **Response to reviewer yKNV (1/3)**
>
> Thank you very much for your thoughtful and constructive review.
>
> ## Explanation on “Lightweight” claim on description-based attribution strategy
>
> In the IMG_RAW setting, each image is represented by **hundreds of patch tokens** in the raw mode (e.g., **256** image tokens for the 16*16 Qwen2.5VL image encoder; **576** image tokens for LLaVA-NeXT);  whereas the caption-as-source mode contributes only **a single caption sentence (averaging 27 tokens among the multimodal samples)**. Since cross-attention is $O(T\_{src}*T\_{gen})$, the image portion of the source sequence is therefore **10-20 times shorter** in the caption mode. This directly reduces:
> * cross-attention FLOPs involving visual tokens;
> * activation memory for storing key/value caches for image tokens;
> * latency for attention-based attribution (we only need to read off attention to caption tokens, not hundreds of patches).
>
> We hope this clarifies that our “lightweight” claim is grounded in the token budget and attention complexity, not just intuition.
>
> ## Attention relevance != causal contribution
> We fully agree that **attention ≠ causation** in general and that naive attention-as-explanation can be misleading. Our goal in this work is more modest:
> * We treat attention as a **practical attribution signal** for *provenance* (i.e., “where did the model look when producing this sentence?”), not as a fully faithful causal explanation.
> * We explicitly **stabilize** this signal via averaging over layers/heads and via top-k + majority voting and thresholding, and then **evaluate** the resulting sentence-level attributions against reference labels.
>
> Recent work Attribute-with-Attention [1], also shows that treating multi-head attention patterns as features, supervised by ablation-based labels, can yield faithful token-level attributions at much lower cost than brute-force ablations. Our work is complementary:
> * [1] **learns** how to combine attention for attribution using additional supervision and model training.
> * Our method is **training-free** and designed specifically for **sentence- and image-level provenance during clinical summarization**, where additional training or ablation runs are often impractical.
>
> ## Key innovations
> Thank you for pushing us to sharpen the novelty discussion. We see three main contributions beyond prior attention-based / retrieval-based attribution work:
>
> ```Generation-time, training-free, multimodal citation pipeline```
>
> Most prior work either:
> * uses **post-hoc** scoring or LLM judges on completed summaries (e.g., DocLens [2]), or
> * trains **surrogate models** or modified attention mechanisms (e.g., AT2 [1]) to approximate causal influence.
>
> In contrast, our framework uses only the native decoder attention of existing MLLMs; runs **during generation (no second pass or retraining)**.
>
> ```Unified treatment of multimodal evidence```
>
> To our knowledge, existing source-identification and attribution methods for summarization are almost entirely **text-only [1, 2]**. We propose:
> * a **raw image** mode that aggregates attention over patch tokens into a single image evidence variable, and
> * a **caption-as-source** mode that replaces images with caption spans, allowing the same machinery to operate purely in text space.
>
> This unified design lets us **compare and quantify trade-offs** between direct visual grounding and text-only surrogates in the same framework
>
> ```Top-k majority + thresholding recipe for sentence-level citations```
>
> Prior attention-based explanations often rely on single-head or layer-specific attention visualizations [3]. We show that moving from token to **sentence-level** citations requires a specific, non-obvious combination:
> * pooling over layers/heads to stabilize attention,
> * taking top-k attended source tokens per output token, majority voting over their sentence IDs, and
> * an aggregation threshold τ at the sentence level.
>
> Our ablations (Fig. 2, Sec. 5.2.1) demonstrate that naive alternatives (e.g., “max” assignment, no threshold) collapse Macro-F1 to near-random levels, while the proposed recipe yields robust performance. We will expand Sec. 3 and Related Work to clearly contrast our approach with (i) AT2-style learned attribution, (ii) retrieval-based similarity methods, and (iii) prior text-only source identification, emphasizing the **generation-time, training-free, multimodal, and sentence-level** aspects that distinguish our work.

---

> > ### Author Response · Authors · 2025-11-21
> > **Response to reviewer yKNV (2/3)**
> >
> > ## Qualitative analysis of attribution errors
> > We agree that understanding failure modes is critical, especially in clinical settings. In the current draft we provide primarily **quantitative** evidence (Macro-F1 and EM) aggregated over datasets. In parallel to the review process, we have begun a **qualitative error analysis** by manually inspecting cases where our attribution disagrees with the reference.
> >
> > Our preliminary observations (which we will document in the appendix) reveal several recurring failure modes:
> > 1. **Merged evidence.** The summarizer often merges information from multiple nearby sentences into one concise sentence, while the reference attribution may mark only a subset of contributing sentences.
> > 2. **Diffuse attention over repeated phrases.** When the same clinical phrase (“shortness of breath”, “no acute findings”) appears multiple times, attention is spread across occurrences, making it ambiguous which specific span should be credited.
> > 3. **Hallucinated or speculative content.** In some cases specific to the IMG_CAP mode, the model introduces mild hallucinations (e.g., inferring a likely diagnosis), and attention focuses on loosely related sentences. Our method still attributes such statements to the “closest” evidence, but this may give a misleading impression of support.
> >
> > In the revision we would clarify that our method is intended as a **decision-support tool** (surfacing candidate evidence) rather than a formal guarantee of correctness, and that misattributions should be interpreted as *signals to inspect* rather than proof.
> >
> > ## Clinical deployment and generalization to other document/image types
> > We appreciate this point and will tone down our deployment claims in the paper.
> >
> > Our current experiments are explicitly limited to **two summarization settings**:
> >  (1) doctor–patient dialogues (CliConSummation), and
> >  (2) chest X-ray Findings → Impression summarization (MIMIC-CXR).
> > We intended these to represent two important, complementary scenarios (multi-turn dialogues and imaging reports), not to claim full coverage of all clinical documentation.
> >
> > We will revise the introduction and conclusion to clarify that we **do not claim immediate clinical deployment**, but rather that:
> > * Our framework is **model-agnostic** and easily pluggable into existing MLLMs,
> > * It provides **transparent provenance signals** that could be valuable to clinicians, and
> > * It is a step toward more accountable clinical summarization systems.
> >
> > Regarding broader applicability, the method itself depends only on **decoder cross-attention tensors** and a **sentence-level segmentation** of the source; it does not rely on chest-X-ray-specific pretraining. It should therefore extend naturally to other document types (e.g., discharge summaries, surgical notes) and imaging modalities (CT, MRI).
> >
> > Following your suggestion, we incorporated **52 additional high-quality samples** from MIMIC-IV-Ext-BHC. This enables stronger cross-dataset assessment. These results show that our method generalizes robustly across datasets and we are actively planning follow-up experiments as future work.
> >
> > | Method            | Macro-F1 | Exact Match |
> > |-------------------|----------|-------------|
> > | **Ours (Qwen2.5-VL)** | **74.30** | **52.16** |
> > | Sent-Embedding    | 73.12    | 43.24       |
> > | Self-Attribution  | 36.42    | 22.43       |

---

> > > ### Author Response · Authors · 2025-11-21
> > > **Response to reviewer yKNV (3/3)**
> > >
> > > ## Framework Component-level ablation and cross-model robustness
> > > We would like to re-iterate the three core steps of our pipeline:
> > > * **Chunking** map input tokens to interpretable evidence units (sentences, plus image span). Without chunking, we cannot obtain sentence-level citations at all, only raw token heatmaps. Chunking is therefore structurally necessary to match our evaluation labels, which are defined at sentence level.
> > > * **Attention pooling & token-level attribution** average attention over layers/heads, take top-k source tokens, majority-vote sentence IDs. Our ablations show that replacing majority voting with max severely degrades performance and also targets different choices of hyperparameters regarding aggregation and de-noising.
> > > * **Sentence-level aggregation** is reversely analytical of chunking
> > >
> > > We agree that robustness across models is important. In addition to Qwen2.5-VL, we ran a full hyperparameter sweep for **LLaVA-NeXT** as well. The optimal setting we found was $k=5, \tau=0.12$. This choice is already reflected in the caption of Table 2, but we did not include the full sweep table for space reasons. We will add a full hyperparameter sweep table for LLaVA-NeXT in the appendix (analogous to the one for Qwen2.5-VL).
> > >
> > > ---
> > >
> > > We hope these clarifications could address your concerns, and we will improve the presentation of this paper in the revision. Please kindly let us know if there are further questions or concerns; otherwise, we would greatly appreciate it if you could consider whether those clarifications warrant a reevaluation of your rating.
> > > [1] Cohen-Wang, Benjamin, Yung-Sung Chuang, and Aleksander Madry. "Learning to Attribute with Attention." arXiv preprint arXiv:2504.13752 (2025).
> > > [2] Xie, Yiqing, et al. "Doclens: Multi-aspect fine-grained medical text evaluation." Proceedings of the 62nd Annual Meeting of the Association for Computational Linguistics (Volume 1: Long Papers). 2024.
> > > [3] Huang, Qidong, et al. "Opera: Alleviating hallucination in multi-modal large language models via over-trust penalty and retrospection-allocation." Proceedings of the IEEE/CVF Conference on Computer Vision and Pattern Recognition. 2024.

---

### Official Review · Reviewer_b7Ae · 2025-10-30

**Soundness:** 2
**Presentation:** 3
**Contribution:** 2
**Rating:** 4
**Confidence:** 3

**Summary:**

Cite While You Generate is a training-free framework for doing sentence-level attribution between generated and prompt text. The authors study their framework in the context of medical summarization where evidence attribution has critical implications for patient safety. The authors detail their approach for handling text or multimodal inputs and demonstrate results in either context. The authors also conduct ablation studies to understand the importance of the hyperparameter choices of their method.

**Strengths:**

- The authors present a well founded, important, and timely topic in source attribution in medical text generation. The idea to use attention weights in a training-free approach is a laudable effort to ground generations using existing LLMs.

- The authors demonstrate empirical results with improvements particularly in an exact match metric across multiple evaluations, a metric which is more difficult than F1 where simpler models are competitive, demonstrating the competitive edge of their approach.

- The method is generally well explained with concise and clear equations, defined symbols, and clear pseudocode.

**Weaknesses:**

- Evaluation datasets are very limited at N=256 for text-only and N=263 for multimodal. Additionally, the multimodal evaluation is confined to only 1 dataset and the text-only to 2 datasets. The task of automated summarization and grounding/attribution in the medical domain has been well studied with many datasets being proposed, this work needs to draw on at least some of those datasets, particularly in the text-only domain:
https://archehr-qa.github.io/
https://biolaysumm.org/#data
https://physionet.org/content/bionlp-workshop-2023-task-1a/1.0.0/
https://vilmedic.app/misc/bionlp23/sharedtask
https://aclanthology.org/2021.acl-srw.30/
https://aclanthology.org/2022.findings-emnlp.286/
https://physionet.org/content/labelled-notes-hospital-course/1.2.0/
https://pubmed.ncbi.nlm.nih.gov/31445245/

- Under the proposed framework, images are attributed as a whole rather than specific image patches. However this is an oversimplification and limits the utility of attribution in, for example, the radiology domain where findings to impression summarization of a chest xray report may benefit from localization of specific observations. Moreover, under the raw multimodal framework, the image is only ever attributed to one summary sentence. Again this is a simplification where complex images may present more than one attributable detail. This is a similar problem with the caption-as-image multimodal framework, where a single caption is used to summarize the image; in many ways, one can consider the findings or impression section of a chest xray to be its caption and in either section, a single sentence rarely describes an entire xray.

- While the ablations of the hyperparameters in Section 5.2.1 are a nice result, this needs to be done on a validation set. Otherwise, this constitutes a form of overfitting and leakage on the evaluation set (line 436-439), leading to inflation of results. Also, while it’s noted on lines 390-394 that Qwen did better and therefore you use it to test other hyperparameters, are there results to show that other hyperparameter combinations don’t do better for LLaVA-NeXT? It seems like the framework is relatively sensitive to such hyperparameters, as in the Qwen case in Figure 2.

- The authors discuss hallucinations as a potential concern of MLLMs (line 57), however their method doesn’t really prevent them. In fact, all it does is provide supporting “evidence” for potentially hallucinated text.

**Questions:**

- The precise model used as the LLM judge for construction of the reference set needs to be provided

- Table 2 LLaVA-NEXT text-only attribution exact match result is incorrectly bolded, should be self attribution (last row)

- Does the reference label for raw images vs image captions differ? Could that explain any of the difference in results between the two? Otherwise, is is a bit surprising that including the raw image tokens allows the model to generate more precise answers (Section 5.2.2)?

---

> ### Author Response · Authors · 2025-11-21
> **Response to reviewer b7Ae**
>
> Thank you very much for your thoughtful and constructive review.
> ## Dataset size and diversity
> We appreciate your concern regarding dataset scale.
>
> Our filtering choices were **intentional and task-specific**, rather than a limitation of data availability. For sentence-level attribution, many clinical datasets (including MIMIC-CXR) contain structural characteristics that break reference alignment: for example, radiology reports referencing prior studies, multi-visit bundles, or impressions covering multiple reports. These create inconsistent or unusable attribution labels. Our strict filtering (Appendix B) ensures high-quality provenance labels. For example, in MIMIC-CXR, we include only cases where the *Impression* summarizes a single *Findings* section.
>
> Following your suggestion, we incorporated **52 additional high-quality text-only samples** from MIMIC-IV-Ext-BHC, bringing us to:
> * 308 text-only samples
> * 571 total samples
> | Method            | Macro-F1 | Exact Match |
> |---------------|----------|-------------|
> | **Ours (Qwen2.5-VL)** | **74.30** | **52.16** |
> | Sent-Embedding    | 73.12    | 43.24       |
> | Self-Attribution  | 36.42    | 22.43       |
>
> This enables stronger cross-dataset assessment. These results show that our method generalizes robustly across datasets and will be included in the revision.
>
> ## Image attribution granularity
> Thank you for raising this important point. Patch-level attention in MLLMs is extremely noisy without grounding supervision. From our experiments on multimodal data:
> Max pooled average attention:
> * text tokens: 0.0212
> * image tokens: 0.0008
>
> We experimented with patch-level normalization (min-max, scaling by patch count, etc.), but none yielded meaningful or stable attributions. Additionally, in MIMIC-CXR, the Findings section already encodes expert localization [1]. Since our goal is **summarization attribution**, not **diagnostic localization**, whole-image attribution aligns with the intended use case. We agree that an image may support several summary sentences. Our next-step extension includes:
> * allowing multiple summary sentences to receive image attribution
> * integrating gradient-based methods (e.g., GradCAM) for spatial localization
> We will clarify this motivation and future direction in the revision.
>
> ## Hyperparameter sweep
> ```Proper validation/test split```
>
> We re-ran the hyperparameter tuning on a **400-sample validation set** and evaluated on the remaining **119 samples**.
> On validation, we see:
> * majority voting again dominates max
> * $k$ = 3 is optimal
> * $\tau$ ≈ 0.15-0.16 yields the best Macro-F1
> These results match our earlier findings and confirm that conclusions are not artifacts of test-set tuning.
>
> ```LLaVA-NeXT hyperparameters```
>
> We would like to clarify that we also ran a full sweep for LLaVA-NeXT and found the optimal setting to be $k = 5, \tau = 0.12$. This is reflected in the caption of Table 2 but not yet fully documented; we will include the complete table in the appendix in addition to the hyperparameter tuning results for the Qwen model. We appreciate you pointing out this omission.
>
> ## Attribution vs hallucination
> We would like to clarify that the goal of this work is traceability, not hallucination suppression. Prior work (DocLens [2]; SelfCheckGPT [3]) shows that:
> * Even when hallucinations occur, clinicians benefit from seeing where claims originate
> * Provenance increases trust and auditability and makes unsupported statements become easier to flag
>
> Because our method is **training-free**, it lays groundwork for future applications such as:
> * Reinforcement signals to guide models away from weakly grounded generations
> * Hallucination detectors based on low grounding frequency
>
> Thanks for pointing this out and we will clarify in the paper that attribution complements hallucination mitigation, rather than replacing it.
>
> ---
>
> We hope these clarifications could address your concerns, and we will improve the presentation of this paper in the revision. Please kindly let us know if there are further questions or concerns; otherwise, we would greatly appreciate it if you could consider whether those clarifications warrant a reevaluation of your rating.
>
> [1] https://github.com/MIT-LCP/mimic-cxr/blob/master/website/content/data/reports.md
>
> [2] Xie, Yiqing, et al. "Doclens: Multi-aspect fine-grained medical text evaluation." Proceedings of the 62nd Annual Meeting of the Association for Computational Linguistics (Volume 1: Long Papers). 2024.
>
> [3] Manakul, Potsawee, Adian Liusie, and Mark Gales. "Selfcheckgpt: Zero-resource black-box hallucination detection for generative large language models." Proceedings of the 2023 conference on empirical methods in natural language processing. 2023.

---

> > ### Comment · Reviewer_b7Ae · 2025-11-27
> >
> > Thank you for the detailed response.
> >
> > In particular, adding 52 additional samples provides additional evidence for your approach.
> >
> > Thank you for your explanation regarding my point: "Under the proposed framework, images are attributed as a whole rather than specific image patches."  We understand your explanation that localizing to a path is future work and that the goal of the paper was "summarization attribution, not diagnostic localization," but I am concerned that is still a limitation of the work, and I'm comfortable with my score.

---

### Meta-Review · Area_Chair_ZmZE · 2025-12-21

**Summary:**

The paper presents a training-free, generation-time source-attribution framework for clinical summarization that leverages decoder attention to cite supporting text spans or images for each generated statement, which avoids post-hoc attribution or model retraining. For multimodal inputs, it supports both raw image-patch attention and a caption-as-span alternative that converts images into captions for text-only alignment. Experiments on two clinical domains demonstrate improvements over embedding-based and self-attribution baselines.

Reviewers raised several recurring concerns: (1) the proposed attribution reflects correlation rather than causality, which may limit suitability for high-stakes clinical use; (2) the novelty relative to prior “attention-as-attribution” work is not sufficiently highlighted; (3) the evaluation scope appears limited, including dataset coverage and ablation depth; (4) image attribution is primarily handled at the image level rather than enabling fine-grained, patch-level grounding; and (5) reliance on LLM-as-a-judge may introduce evaluation bias.

The rebuttal addresses some of these points to a degree, but key issues remain and not all reviewers were fully convinced. The authors are encouraged to incorporate this feedback to strengthen the positioning, evidence, and analysis in a future submission.

**Reviewer Concerns:**

The authors’ rebuttal attempts to address the main concerns raised by the reviewers. It also acknowledges that image attribution is largely handled at the whole-image level, which remains a limitation. However, the remaining issues would still require substantial revisions to be addressed more thoroughly in the paper.

**Reviewer Scores:**

Given the original reviews and the authors’ rebuttal, it seems likely that the reviewers will maintain their original scores.

---

### Decision · Program_Chairs · 2026-01-26

Reject